# On the Simplicity-Similarity Tradeoff of LoRA and Full Fine-Tuning

**Jerome Emery[1]  Darshan Patil[2,3,4]  François Leduc-Primeau[1]  Sarath Chandar[2,3,4,5]
Ekaterina Lobacheva[2,3,4]**
[1]Polytechnique Montréal    [2]Chandar Research Lab    [3]Mila – Quebec AI Institute
[4]Université de Montréal    [5]Canada CIFAR AI Chair
Correspondence: `jerome-2.emery@etud.polymtl.ca`

## Abstract

Fine-tuning is the dominant paradigm for adapting pre-trained models to downstream tasks. However, mounting evidence suggests that parameter-efficient methods, such as *Low-Rank Adaptation (LoRA)*, converge to distinct solutions compared to *Full Fine-Tuning (FFT)*. In this work, we investigate the underlying optimization biases driving this divergence. We demonstrate a clear shift in their learning dynamics: FFT exhibits a strong *simplicity bias*, regardless of the downstream tasks. LoRA, meanwhile, consistently prioritizes features already prevalent in the pre-training distribution, a phenomenon which we term *similarity bias*. Our findings provide a feature-level explanation for observed differences between LoRA and FFT, offering potential critical insights into how adaptation strategies influence model robustness and task-specific generalization.

## 1  Introduction

The paradigm of adapting pretrained models through finetuning has enabled the creation of extremely capable models on specialized downstream applications. As models scale, however, full fine-tuning (FFT), which updates the entire set of network parameters, became increasingly prohibitive, resulting in the development of parameter efficient training methods. Low-Rank Adaptation (LoRA) enables efficient adaptation to downstream tasks by constraining the weight update to a low-rank decomposition Hu et al. (2022).

Recent work has investigated the extent to which LoRA and FFT differ in practice. Notably, Biderman et al. (2024) show that LoRA underperforms FFT, but also better maintains the base model's performance. While this could potentially be attributed to LoRA learning at a different rate or having less capacity compared to FFT, Shuttleworth et al. (2025) find structural differences in the changes made to the pretraining weights by LoRA and by FFT, even when trained to equal performance on the downstream task. This suggests that LoRA might in fact be learning a different solution altogether, relative to FFT.

Neural network solutions are driven by inductive biases that arise from architecture, optimizer, and data statistics (Pascanu et al., 2025). Among these, simplicity bias, the tendency of neural networks to rely on simpler features over more complex but equally predictive ones, has been extensively documented in models trained from scratch (Shah et al., 2020; Hermann & Lampinen, 2020; Chen et al., 2021; Geirhos et al., 2019). In the transfer learning paradigm, however, we need to also account for the inductive bias induced by the model's pretrained initialization on the learning dynamics (Pascanu et al., 2025). For example, Trivedi et al. (2023) show that FFT is more susceptible to simplicity bias compared to linear probing on pretrained models; they do not explore how the downstream features and their relation to the pretraining distribution affect the resulting models.

In this work, we investigate the differences in the learned solutions of LoRA and FFT through a feature reliance analysis. We first assemble a collection of image classification tasks, considering each as being representative of a "feature". Following Shah et al. (2020), we finetune the models on paired adaptation tasks created by concatenating the samples of two datasets into one input (Fig. 4). By evaluating models on dataset variants where one feature is shuffled, we can then probe the model's reliance on the other feature.

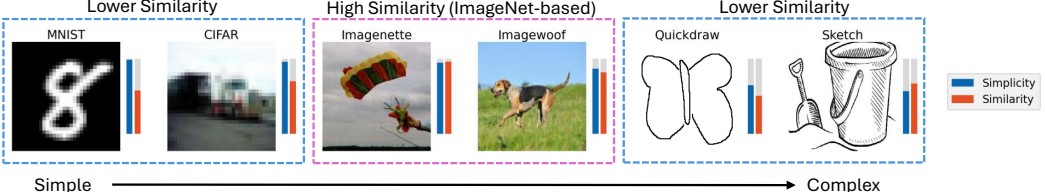

Figure 1: Visualization of the six base datasets, ranked by simplicity.

Overall, we see that FFT solutions are characterized by a simplicity bias while LoRA solutions favor features closer to the pretraining distribution. We term the latter similarity bias.

This results in three distinct regimes of feature balance for FFT and LoRA: (1) When the features differ in similarity and the more similar feature is also the simpler, FFT and LoRA solutions behave similarly, relying on the simpler/similar feature. (2) When the two features are approximately the same similarity, LoRA learns a more balanced feature representation while FFT heavily favors the simpler feature. (3) When the features differ in similarity and the more similar feature is harder to learn, FFT prefers the simpler feature (resulting in a balanced feature representation), but LoRA heavily prefers the more similar feature.

## 2 METHODOLOGY

### 2.1 FEATURE SIMPLICITY AND SIMILARITY

In the context of fine-tuning, both feature simplicity and similarity to pretraining are confounded. Figure 1 shows the six base datasets we chose to cover a wide range of simplicity and similarity to the pretraining distribution. To measure the simplicity and similarity of these datasets, we first train a model on each individually, and define the *simplicity* of a feature as the maximum validation accuracy achieved by FFT $\text{Acc}_{\text{FFT}}$. The similarity to pretraining, which we shorten to *similarity*, is defined as the ratio of linear probing accuracy to FFT accuracy $\frac{\text{Acc}_{\text{LP}}(\mathcal{D}_i)}{\text{Acc}_{\text{FFT}}(\mathcal{D}_i)}$. The rankings with respect to each property are discussed in greater detail in Appendix B.6.

### 2.2 ADAPTATION SETTINGS

**Multi-Feature Datasets** We construct two-feature datasets by combining pairs of image classification datasets into a single task. Each input $x = (x_A, x_B)$ pairs two images drawn from distinct source datasets, paired such that they share the same label $y \in \{1, \ldots, 10\}$. Both features are independently predictive of $y$. Each dataset is comprised of 10 classes.

**Feature Reliance** Throughout our experiments, we quantify feature reliance by evaluating fine-tuned models on dataset variants where one feature is shuffled (replaced with a random image sampled from the same dataset) while the other remains correlated with the label. We refer to the feature reliance of $x_A$ (resp. $x_B$) as the validation accuracy on a dataset where $x_B$ (resp. $x_A$) is randomized.

**Model Architecture** All experiments use a Vision Transformer Base (ViT-B) Wu et al. (2020) backbone pretrained on ImageNet-21k Deng et al. (2009). We compare three adaptation methods: linear probing (frozen backbone), FFT, and LoRA with ranks $r = \{1, 4, 16, 64\}$. The full experimental details are reported in Appendix B.

## 3 RESULTS

We fine-tune models on all 15 feature pairs formed from the six source domains, evaluating linear probing (LP), LoRA, and FFT. Figure 2 summarizes our findings. For each feature pair, we plot the difference in feature reliance between $x_A$ and $x_B$, where $x_A$ is the simpler feature of the pair.

We find that the interaction between simplicity bias and similarity bias leads to different behaviors depending on the properties of each feature pair. We analyze these differences by grouping the

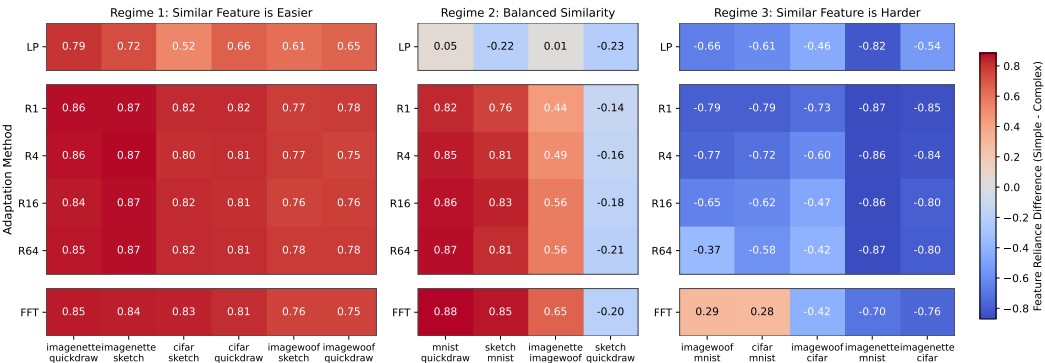

Figure 2: Evolution of feature balance across adaptation methods. Each column represents a dataset pair, with rows showing the progression from linear probing (LP) through LoRA ranks to FFT. Warmer colors indicate stronger reliance on the simple feature.

feature pairings into three adaptation regimes. **Regime 1** contains pairs where the more similar feature is also the simpler one. **Regime 2** includes pairs where both features have comparable similarity to pretraining, and **Regime 3** contains pairs where the more similar feature is also more complex.

**Regime 1: Similarity and Simplicity Aligned**   The first regime contains settings where similarity and simplicity bias are aligned towards the same feature. Figure 2 (left) shows that LoRA and FFT exhibit near identical feature reliance, both solving the task via the simple/similar feature. We illustrate this further in Figure 3 (column 1) which shows the evolution of validation accuracy, as well as simple and complex feature reliances for the *CIFAR-Quickdraw* experiment which frames *CIFAR* as the more simple and similar feature of the pair. While the complex feature reliance stays at random accuracy, the simple feature reliance matches the validation accuracy almost perfectly, suggesting that all methods learn the task using the simple feature. Thus, this regime represents a setting where the choice of adaptation method has minimal effect on feature selection.

**Regime 2: Balanced Similarity**   We next investigate the regime where the linear probing baseline learns an approximately equal balance of features, i.e. when the two features are balanced in similarity. Looking at the simple feature reliance of LoRA and FFT in Figure 2 (middle), we begin to observe differences between LoRA and FFT. Across most pairings, FFT heavily relies on the simpler feature. LoRA also relies on the simpler feature, but this is modulated by the rank, with the lower rank solutions relying less on the simpler feature than the higher rank solutions. Looking at the representative training dynamics curves in Figure 3 (column 2), we see that all models do initially start to learn both features. The complex feature, however, eventually starts being downweighted; this effect starts earlier and at a lower complex feature peak reliance for FFT, resulting in the FFT solution being significantly more biased towards the simple feature. LoRA undergoes this downweighting effect later in training and at a higher peak, learning to use both features to improve accuracy, before shifting to the simpler feature. In this regime, while both methods are susceptible to simplicity bias, LoRA maintains better feature balance between simple and complex features.

One exception to this pattern is the *sketch-quickdraw* pairing, which we discuss below.

**Regime 3: Similar Feature is Complex**   We now look at the regime where the more similar feature is the more complex feature, i.e. where the similarity bias and simplicity bias are in most direct opposition. Here, we see the greatest difference in feature reliance between the FFT and LoRA, with FFT relying comparatively much more on the simpler feature, and LoRA relying much more on the similar feature. This effect is stronger the lower the LoRA rank. The training dynamics for this regime (Figure 3, column 3) illustrate the drastic differences in feature reliance evolution. Once FFT starts learning the simpler feature, its reliance on the complex, similar feature immediately plateaus. LoRA meanwhile, almost exclusively learns on the similar feature, with only a slight increase in reliance on the simpler feature near the end of training.

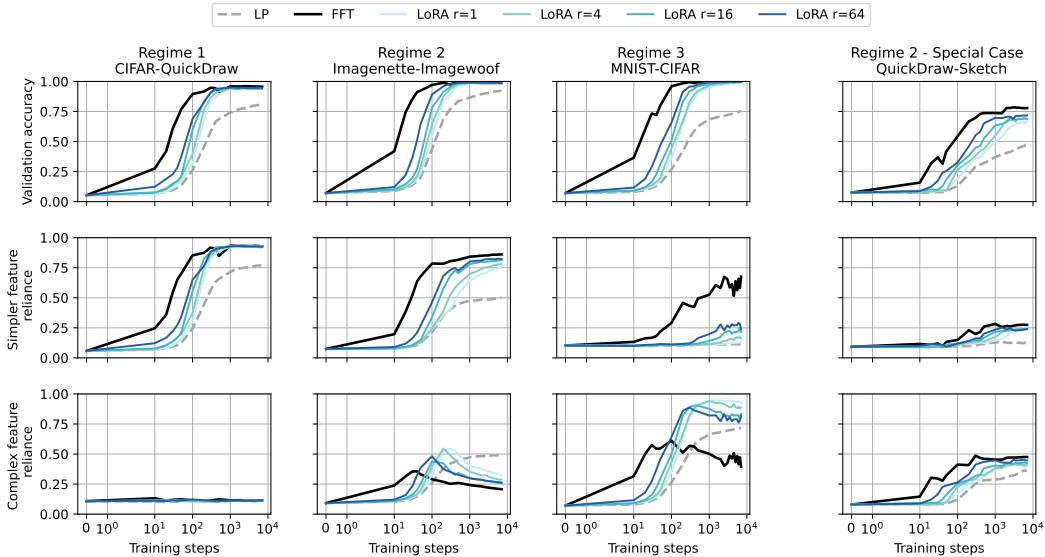

Figure 3: Evolution of validation accuracy, simple feature reliance and complex feature reliance over training steps. Each column shows a pairing from each regime, with the last pairing shown as a special case.

The observed dynamics in this regime provide a potential explanation for LoRA forgetting less than FFT at equal performance. The simplicity bias of FFT causes it to downweight features aligned with pretraining in favor of simple features predictive of the downstream task. LoRA achieves comparable accuracy through a feature selection driven by similarity bias.

**When LoRA Learns Less: Sketch-Quickdraw**   We conclude with the special case mentioned above of the *sketch-quickdraw* pairing. This setup presents the case where LoRA does not have sufficient capacity to learn both features, connecting to prior findings that LoRA underperforms FFT on difficult downstream tasks. The last column of Figure 3 shows the *sketch-quickdraw* pairing. *Sketch* and *Quickdraw* are the two most difficult features from the feature set. For this pairing, the difference is not in which features are learned, but rather in how well they are learned. We can see from Figure 2 that LoRA and FFT learn a very similar feature balance, yet reach different final validation accuracies. As the LoRA rank decreases, the clean validation accuracy also drops. Notably, the feature balance remains fairly stable relative to FFT. This suggests that when both features are sufficiently difficult, LoRA's limited capacity constrains the quality of learned representations rather than the selection of features.

## 4   CONCLUSION

In this paper, we investigate how LoRA and full fine-tuning (FFT) shape feature reliance during transfer learning through two competing inductive biases: feature simplicity and similarity to the pre-training distribution. We find that FFT is generally governed by a simplicity bias, while LoRA is driven by a similarity bias, favoring solutions that leverage features proximal to the pre-training distribution. By grouping feature pairings according to their simplicity-similarity tradeoffs, we identify three adaptation regimes that characterize the functional divergence between LoRA and FFT.

Our work expands on the conclusions of (Biderman et al., 2024), that "LoRA learns less and forgets less," by exploring how this behavior influences the specific features selected and developed during adaptation. We demonstrate that the choice between LoRA and FFT constitutes a strategic selection between competing optimization biases. For example, while FFT's simplicity bias may uncover general solutions, it also risks discarding robust pre-trained features in favor of low-complexity ones. In contrast, LoRA's similarity bias anchors the model to the pre-training distribution, a dynamic that may preserve existing robustness but can also inhibit the acquisition of novel, task-specific features.

## ACKNOWLEDGMENTS

We would like to thank Mila (`mila.quebec`) and its IDT team for providing and supporting the computing resources used in this work. Sarath Chandar is supported by the Canada CIFAR AI Chairs program, the Canada Research Chair in Lifelong Machine Learning, and the NSERC Discovery Grant. Ekaterina Lobacheva is supported by IVADO and the Canada First Research Excellence Fund.

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

## A  RELATED WORK

**LoRA.**  Low-Rank Adaptation (LoRA) (Hu et al., 2022) is a parameter-efficient fine-tuning method that freezes the pretrained weights and learns a low-rank weight update. Despite its practical success, subsequent work has revealed important differences between LoRA and full fine-tuning. Biderman et al. (2024) showed that LoRA underperforms full fine-tuning on challenging target domains such as code and mathematics, but better preserves the base model's capabilities on out-of-distribution tasks. Shuttleworth et al. (2025) identify structural differences between LoRA and full fine-tuning updates and link the spectral differences to the learning-forgetting tradeoff between LoRA and full fine-tuning. Notably, even when LoRA and full fine-tuning achieve comparable accuracy, their solutions occupy structurally different regions of the parameter space, hinting at the fact that both methods may learn different features.

**Simplicity Bias.**  Simplicity bias, the tendency of neural networks to fit the training data with simple functions, has been widely regarded as a key driver of generalization. Many works view simplicity bias as an artifact of gradient descent (Arpit et al., 2017; Lyu et al., 2021; Nakkiran et al., 2019), although recent works have also looked at the role of architecture, initialization, and activation function (Teney et al., 2024; 2025; Goldblum et al., 2024; Valle-Perez et al., 2019). Closely related, Rahaman et al. (2019) study spectral bias, according to which networks learn low-frequency components of the target function. On average, simplicity bias is beneficial for generalization, particularly for limiting overfitting of overparameterized networks. However, simplicity bias can also compromise model robustness. When multiple features are predictive of a label, networks preferentially rely on the simpler one, ignoring other equally predictive features (Shah et al., 2020; Hermann & Lampinen, 2020; Scimeca et al., 2022; Morwani et al., 2023). This phenomenon is often referred to in practice as shortcut learning. In vision domains, it has been shown that models can rely on features like image texture (Geirhos et al., 2019) or background (Beery et al., 2018; Sagawa* et al., 2020). Contrary to our work, simplicity bias is generally studied with models that are trained from scratch, ignoring potential confounding biases that come from fine-tuning pretrained models.

**Feature Distortion in Fine-tuning.**  When fine-tuned on a downstream task, foundation models reuse features learned during pretraining (Neyshabur et al., 2020; Raghu et al., 2019). However the adaptation protocol can influence which features are transferred. Kumar et al. (2022) show that full fine-tuning distorts the pretrained features to improve in-distribution accuracy, which degrades out-of-distribution performance. Andreassen et al. (2022) further show that pretrained models exhibit strong out-of-distribution robustness early in fine-tuning, which vanishes as training proceeds. These two works offer evidence of a tradeoff between learning task-specific features and maintaining representations from the backbone. To mitigate these issues, Kirichenko et al. (2023) show that retraining only the classification head is often sufficient to recover robustness, and Lee et al. (2023) show that selectively fine-tuning specific layers can outperform full fine-tuning. Closest to our work, Trivedi et al. (2023) formalize simplicity bias in the context of adaptation, showing that full fine-tuning is more susceptible to simplicity bias than linear probing. However they do not take into account how the downstream features and their relation to the pretraining distribution affect the resulting models. We continue this line of work by introducing a second axis, feature similarity to the pretraining distribution, and show that LoRA and full fine-tuning learn qualitatively different solutions shaped by the interplay between simplicity and similarity.

## B  EXPERIMENTAL DETAILS

### B.1  MODEL ARCHITECTURE

In all our experiments, we use a Vision Transformer Base (ViT-B) Wu et al. (2020) backbone pretrained on ImageNet-21k Deng et al. (2009) loaded from https://huggingface.co/google/vit-base-patch16-224. We replace the final classification head to match the number of classes in our setup.

### B.2  ADAPTATION METHODS

Linear probing refers to adapting only the classification head, while leaving the backbone model frozen. Full fine-tuning updates all the model parameters. LoRA decomposes the weight update

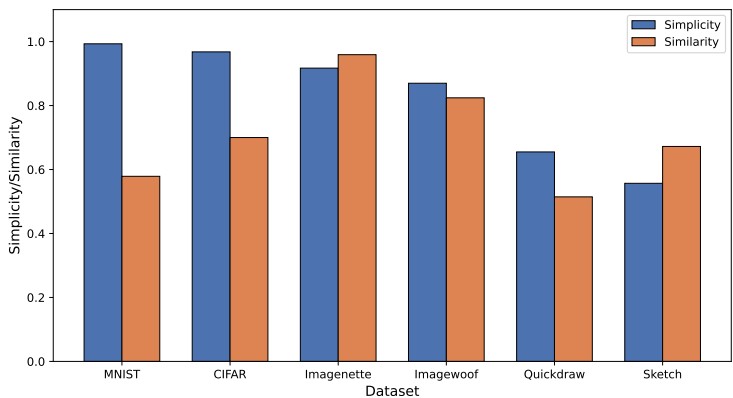

Figure 5: Simplicity and Similarity of each base dataset

$\Delta W$ as $\frac{\alpha}{r}BA$, where $B \in \mathbb{R}^{d \times r}$ and $A \in \mathbb{R}^{r \times k}$ are trainable matrices. The hyperparameter $\alpha$ is chosen to weight the LoRA update. Consistent with the convention used in Biderman et al. (2024), we set $\alpha = 2r$.

### B.3 LoRA Configuration

We use the popular PEFT library Mangrulkar et al. (2022) to fine-tune models with LoRA. We add LoRA adapters to the query, key, and value matrices in the attention layers, and to all dense layers in the MLP blocks. The classification head is trained from scratch jointly with the feature extractor. Layernorm parameters are also kept trainable.

### B.4 Multi-Feature Datasets

To construct the paired feature datasets, we concatenate the samples from two base datasets $\mathcal{D}_A$ and $\mathcal{D}_B$. For each class $c$, we select $n = \min(|\mathcal{D}_A^c|, |\mathcal{D}_B^c|)$ samples, which corresponds to the minimum number of samples of class $c$ in either dataset. Each image is resized to $224 \times 224$. The images are then concatenated along the width, across all 3 channels. The resulting concatenated images have size $3 \times 224 \times 448$. To concatenate grayscale datasets with color ones, we convert them to RGB by replicating the channel three times. Each paired feature dataset uses 10 classes.

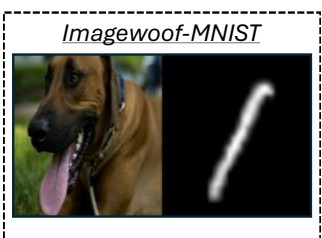

Figure 4: Example of a sample from the Imagewoof-MNIST paired dataset

Once the two-feature datasets are formed, we split the samples into a training and validation set. We use the validation set when evaluating the feature reliance metric described in Section 2.2.

### B.5 Training Procedure

We fine-tune the models using the Adam optimizer with batch size 32. We do not use weight-decay or data augmentations. We train with learning rates $\{10^{-4}, 5 \times 10^{-5}, 10^{-5}\}$. The results reported in the main text are for the intermediate learning rate $5 \times 10^{-5}$. Section C shows the results for the other two.

### B.6 Feature Simplicity and Similarity

Figure 5 shows the simplicity and similarity measure for each feature, as defined in Section 2.1. Note that for the purposes of the single-feature metrics, we train with a single learning rate of $10^{-5}$.

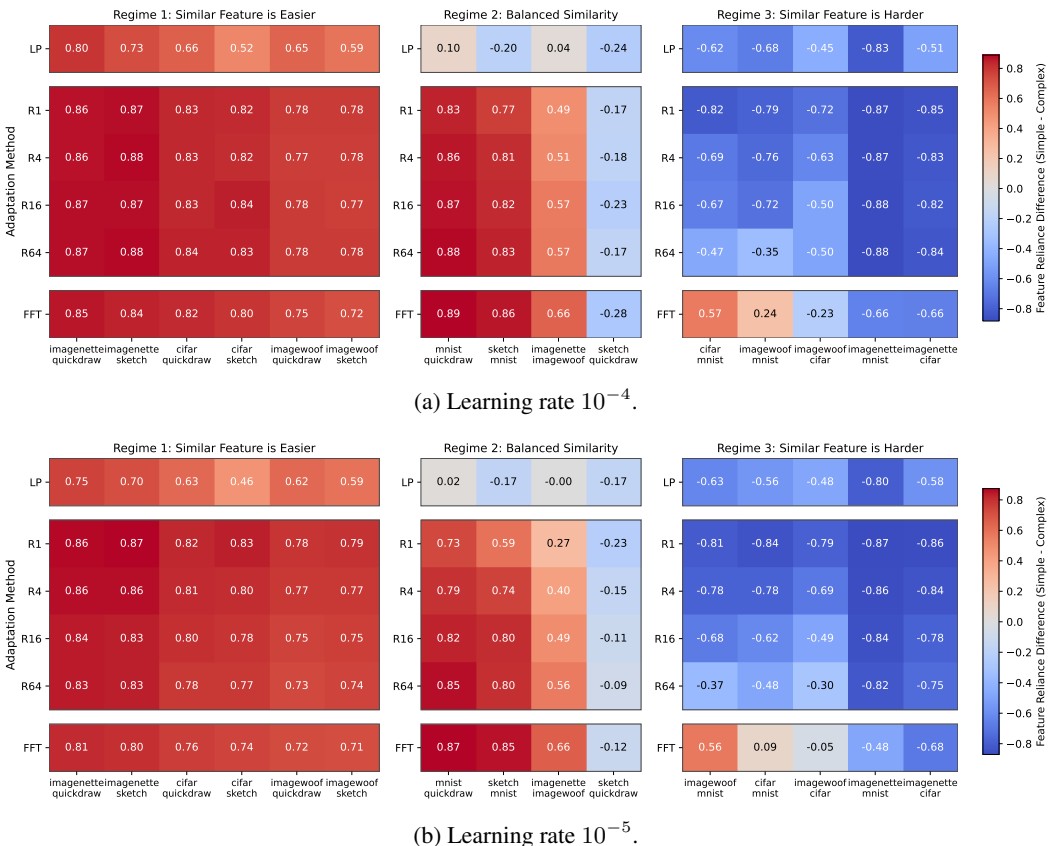

(a) Learning rate $10^{-4}$.

(b) Learning rate $10^{-5}$.

Figure 6: Evolution of feature balance across adaptation methods. Each column represents a dataset pair, with rows showing the progression from linear probing (LP) through LoRA ranks to FFT. Warmer colors indicate stronger reliance on the simple feature.

We find that MNIST (Deng, 2012) and CIFAR (Krizhevsky & Hinton, 2009) are the simplest tasks, followed closely by Imagenette (Howard, 2019a) - a subset of easily classified ImageNet classes. Imagewoof (Howard, 2019b), a fine-grained classification task consisting of dog breeds is harder than Imagenette. The most difficult tasks are the Sketch and Quickdraw domains from DomainNet (Peng et al., 2019).

In terms of similarity, Imagenette and Imagewoof have the highest similarity, as they are subsets of the pretraining dataset. Although CIFAR images resemble the natural images found in the pretraining dataset ImageNet, our similarity analysis suggests that it has significantly less similarity than Imagenette and Imagewoof. This result may be due to the lower resolution of CIFAR. The datasets MNIST, Sketch and Quickdraw have the lowest similarity.

## C  TESTING DIFFERENT LEARNING RATES

We show the final feature reliance results for the two other tested learning rates. In general, the learning rate seems to have little effect on the dynamics described, although with the smallest learning rate $10^{-5}$, certain values of feature reliance are lower. This can be attributed to underfitting compared to the larger learning rates.

We also observe, for all learning rates, that simple feature reliance drops slightly with capacity across certain regime 1 pairs. This behavior seems to arise from overfitting, since the validation accuracy for these pairs also drops accordingly.

