# OpenReview forum: "On the Simplicity-Similarity Tradeoff of LoRA and Full Fine-Tuning"
_ICLR.cc/2026/Workshop/Sci4DL — Sci4DL 2026_

### Official Review · Reviewer_SBo5 · 2026-02-22

**Fit:** 3
**Significance:** 2
**Confidence:** 2

**Summary:**

This paper shows empirically that LoRA and Full Fine-tuning has different implicit biases, where LoRA exhibits similarity bias, and Full Fine-tuning exhibits simplicity bias.

**Strengths:**

The paper is clearly presented and easy to read, the experiment can clearly show the difference between LoRA and full finetuning in different regimes.

**Suggestions:**

In figure (1), it is better to have pictures that share the same class, as said in description of "multi-feature dataset" in section 2.2.

In Figure(2) a, write something clearer than Dataset A - Dataset B, maybe can label them in terms of regimes, Figure 3 it is not clear to have a regime 2 in the fourth column after the regime 3.

It is better to add theoretical analysis to understand the implicit bias driven by these two dynamics, similar to [1].

Gradient dynamics for low-rank fine-tuning beyond kernels. Arif Kerem Dayi, Sitan Chen.

---

### Official Review · Reviewer_Q3vj · 2026-02-26

**Fit:** 3
**Significance:** 3
**Confidence:** 3

**Summary:**

This work examines two fine-tuning paradigms: full fine-tuning (FFT) and Low-Rank Adaptation (LoRA). While FFT is known to outperform LoRA in terms of accuracy, LoRA remains computationally more efficient. However, the inductive biases of these methods have received little attention. This study addresses this gap by investigating two key biases in LoRA and FFT:
- Simplicity bias: the tendency of a model to rely on simpler features over more complex yet equally predictive ones.
- Similarity bias: the tendency of a model to rely on features that align more closely with the pretraining data.

The experimental framework involves fine-tuning a Vision Transformer Base model on paired adaptation tasks. The results reveal that LoRA is primarily driven by a similarity bias, whereas FFT exhibits a stronger simplicity bias.

**Strengths:**

The paper is well-written and tackles an important question for the fine-tuning community. The experimental setup is both clear and relevant.

**Suggestions:**

Comments about readability and typos:
- Line 46: for non specialists, it would be helpful to add a brief sentence explaining what linear probing is
- It was not clear to me until Section 2 that the task was classification, that could be specified in the Introduction (e.g., line 52)
- For clarity, feature balance should be explicitly defined in the caption of Figure 2
- Line 161: typo "it's reliance"

Other comments and questions:
- Interestingly, it appears that the drop in the complex feature reliance only occurs once validation accuracy goes above a certain threshold: this pattern is visible in Figure 2 (regimes 2 and 3) for both FFT and LoRA. If the authors agree with this observation (which could probably be made more quantitative), it might be worth highlighting in the paper, as it unifies the conclusions for LoRA and FFT across different regimes, including Sketch-Quickdraw. I wonder if this might even reflect a general phenomenon, independent of the fine-tuning technique.
- In the same line: on Figure 3, Regime 3, a similar phenomenon occurs for FFT and LoRA (both simple and complex feature reliances increase, then complex feature reliance starts to decrease), but the decrease occurs later with LoRA. Have the authors explored doing more training steps for LoRA, to see if it eventually converges to the same feature balance as FFT—while maintaining validation accuracy?
- Can the authors justify why the simplicity of a feature is defined as *FFT*'s maximum validation accuracy? I wonder whether changing FFT to LoRA in the definition might significantly change the simplicity of features (e.g., simple features could become complex and vice versa). If so, this could potentially impact the conclusions of the paper.

---

### Official Review · Reviewer_vaRE · 2026-02-27

**Fit:** 2
**Significance:** 2
**Confidence:** 1

**Summary:**

This paper studies why LoRA and full fine-tuning (FFT) often converge to different solutions even when they can reach similar downstream performance.  The authors propose that the divergence is driven by two competing optimization biases: Simplicity bias (dominant in FFT): the tendency to rely on “simpler” features; Similarity bias (dominant in LoRA): the tendency to rely on features more prevalent/compatible with the pretraining distribution.

**Strengths:**

A strength of this work is its relevance to how practitioners choose between LoRA and full-parameter fine-tuning. By characterizing when LoRA tends to anchor adaptation to features already supported by the pretraining distribution versus when FFT is more prone to discover simpler downstream shortcuts, the paper provides an actionable lens for anticipating trade-offs in robustness, transfer, and retention under domain shift.  This framing is especially timely given the widespread use of LoRA as a default parameter-efficient adaptation strategy.

**Suggestions:**

To strengthen the generality of the conclusions, broaden the empirical scope beyond a single ViT-B pretrained on ImageNet-21k by testing additional backbones and pretraining sources to verify that the observed LoRA–FFT dynamics are not architecture- or dataset-specific.  Also, more thoroughly validate the proxy definitions of simplicity and similarity. Finally, improve statistical rigor by running multiple seeds and reporting variance (confidence intervals/error bars).

---

### Meta-Review · Area_Chair_wQqZ · 2026-03-01

**Recommendation:** Accept

**Metareview:**

Recommending acceptance. Would note that this misses a lot of related work, and strongly suggest the authors to take a look at the literature to better situate their claims.

---

### Decision · Program_Chairs · 2026-03-02

Accept